# The impact of clade B lineage 5 MERS coronaviruses spike mutations from 2015 to 2023 on virus entry and replication competence

Ray T. Y. So[1], Kenrie P. Y. Hui[1,2], John C. W. Ho[2], Kaman K. M. Lau[1], Ziqi Zhou[3], Michael C. W. Chan[1,2], Leo L. M. Poon[1,2,4], Malik Peiris[1,2,4]*

1 School of Public Health, LKS Faculty of Medicine, The University of Hong Kong, Hong Kong SAR, China, 2 Centre for Immunology & Infection, Hong Kong Science Park, Hong Kong SAR, China, 3 Centre of Immunobiology and Infection, Blizard Institute, Queen Mary University of London, London, United Kingdom, 4 HKU-Pasteur Research Pole, School of Public Health, LKS Faculty of Medicine, The University of Hong Kong, Hong Kong SAR, China

* malik@hku.hk

## Abstract

Middle East respiratory syndrome coronavirus (MERS-CoV) is an emerging coronavirus that can cause zoonotic disease in humans with lethal severe viral pneumonia. Dromedary camels are the source of zoonotic infection. As of November 2025, MERS-CoV has resulted in a total of 2630 reported cases, 37% of these being fatal. The number of reported human cases has been on a decreasing trend since 2016 and reached a nadir during the COVID-19 pandemic. The reason for the reduction of cases is unclear and may be multifactorial. We hypothesized that mutations accumulating in the virus spike protein may have reduced zoonotic potential. Here, we investigate the impact of recently emerged virus spike-protein mutations on virus replication competence using pseudoviruses and replication-competent recombinant viruses. We found that virus spike variants detected in 2019 and some from 2023 show a reduced cell entry, lower viral replication and reduced fitness in human primary alveolar epithelial cells and multiple cell lines. All the MERS-CoV spikes tested showed a cell-entry pathway preference via the cell-surface TMPRSS2 route. Mechanistically, we showed the V530A mutation in the 2019 spike sequence had a reduced human DPP4 binding phenotype. Our data highlighted MERS-CoV spike mutations can modulate viral fitness in human cells and provide new insights to understand recent MERS epidemiology.

### Author summary

MERS-CoV is identified by the World Health Organization (WHO) as a potential pandemic candidate. The ability of coronaviruses to mutate and adapt in new hosts raises concerns about the impact of virus genetic changes on human

**Data availability statement:** NGS sequencing data and data points for figures have been deposited to https://github.com/RScode23/spike-study.

**Funding:** The research was fully supported by a grant from the Research Grants Council of the Hong Kong Special Administrative Region, China (Project No. [T11-705/21-N]; https://www.ugc.edu.hk/eng/rgc/) to M.P.. This study was supported in part by the InnoHK initiative of the Innovation and Technology Commission of the Hong Kong Special Administrative Region Government (Health@InnoHK: C2i; https://www.innohk.gov.hk/en/r-d-centres/health-innohk/) to K.H., J.H., M.C. L.P. and M.P.. The funders had no role in study design, data collection and analysis, decision to publish, or preparation of the manuscript.

**Competing interests:** The authors have declared that no competing interests exist.

zoonotic and pandemic potential. There has been a notable decline in human MERS cases reported to the WHO since 2018, but the underlying reasons remain unclear. Here, we focus on investigating whether the recently emerged virus spike mutations may contribute to this observation. We found that some spike mutations detected in 2019 and 2023 viruses showed reduced cell entry, viral replication and viral fitness, of variable magnitudes. This study highlighted a need for comprehensive genomic surveillance and phenotyping of recent MERS-CoV isolates, in particular mutations at the spike protein, to understand its emergence potential.

## Introduction

Middle East respiratory syndrome coronavirus (MERS-CoV) is a highly pathogenic zoonotic coronavirus that was first reported in a patient with severe viral pneumonia in Saudi Arabia in 2012 [1]. Dromedary camels are recognized as a major source of zoonotic infection [2]. Surveillance studies subsequently revealed MERS-CoV circulation in dromedary camels in the Arabian Peninsula, Northern, Eastern and Western Africa, and in the Central Asia, e.g., Pakistan [3–11]. While camel exposure was associated with an increased risk of human disease, the exact camel-to-human transmission mechanisms remain unclear. As of 20th November 2025, MERS-CoV has resulted in a total of 2630 reported human cases, causing 963 deaths with a fatality rate of 37% (information from WHO [12]). These reported cases include primary zoonotic infections, as well as clusters of human-to-human transmissions in hospital settings, as documented in Saudi Arabia [13,14]. In 2015, a single returning traveler from the Arabian Peninsula to the Republic of Korea initiated an outbreak resulting in a total of 186 confirmed cases [15]. Increased awareness and improved hospital infection prevention and control measures have contributed to the reduction of MERS cases and deaths since 2016 [16]. Interestingly, since 2017, there was a further reduction of human cases reported from the Arabian Peninsula (WHO statistics, Fig 1A). The trend of reduced human cases has persisted even after the relaxation of public health and social restriction measures implemented during the COVID-19 pandemic. The reason for the reduced case reports of MERS may be multifactorial, including better infection control measures in hospitals, increased public awareness to coronavirus diseases, cross-protection from COVID vaccination or infection, or changes in the MERS-CoV phenotype. In this study, we investigate the potential impact of recently emerged mutations in the virus spike protein on virus replication competence, as one potential explanation.

The genetic evolution of MERS-CoV primarily occurs within the camel hosts [17]. Multiple lineages of MERS-CoVs have been shown to circulate among camels, which later underwent genetic recombination to result in a recombinant lineage (clade B lineage 5), which was associated with large outbreaks in 2015 [6]. Experimental data showed the clade B lineage 5 had a higher viral replication fitness in vitro and ex vivo

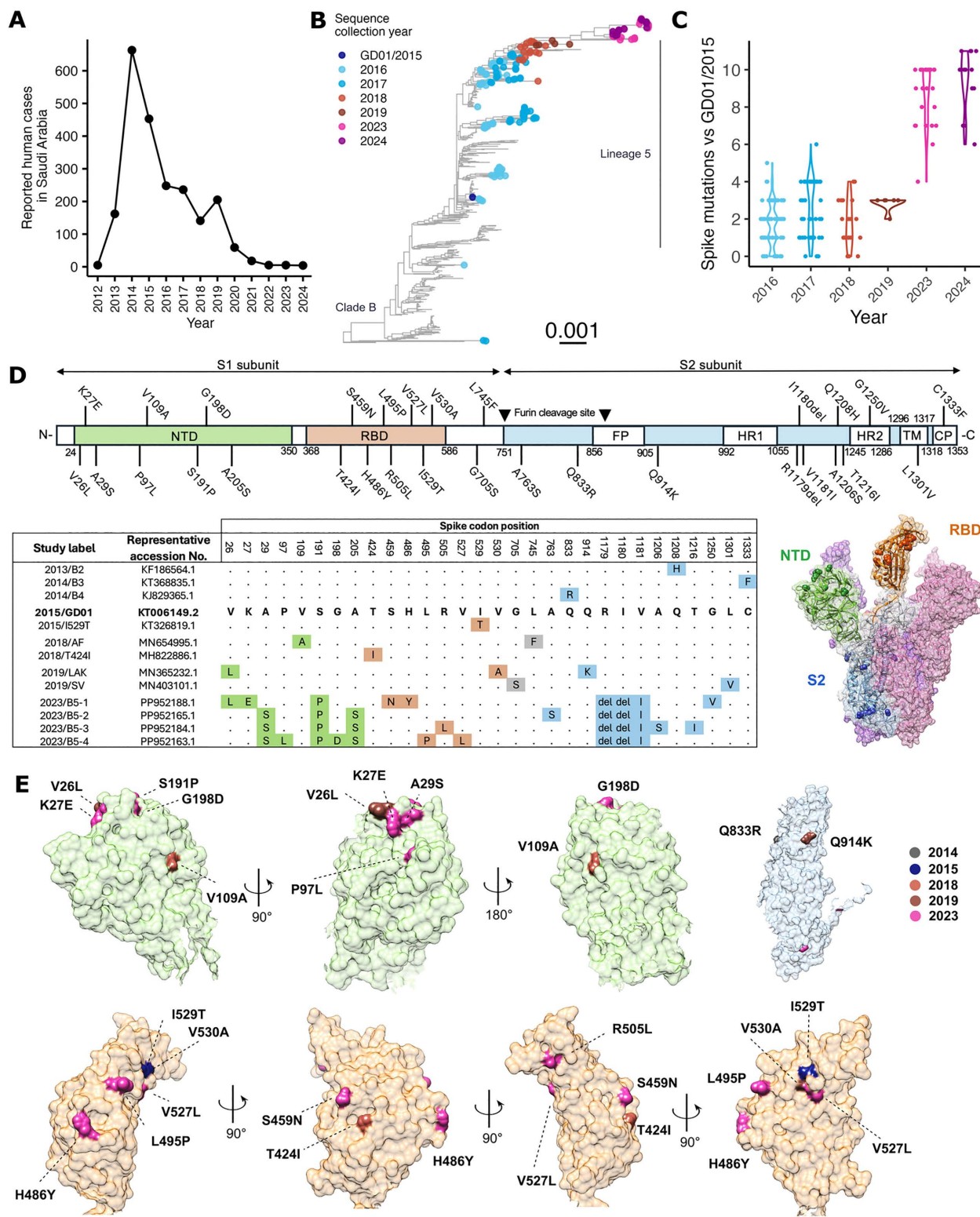

**Fig 1. MERS-CoV human cases and the accumulation of spike amino acid mutations. (A)** Numbers of lab confirmed human MERS-CoV cases in Saudi Arabia reported to the World Health Organization (WHO) [12]. **(B)** Full genome ML tree of 607 MERS-CoV sequences. Reference sequence MERS-CoV/GD01 collected in 2015 and sequences with collection year in 2016 and later are colored. Scale bar represent nucleotide change per site.

**(C)** Dot and violin plot of number of spike amino acid mutation from sequences collected from 2016 to 2024 against the reference sequence 2015/GD01. **(D)** Illustration showing the domains of MERS-CoV spike protein and the position of spike mutations tested in this study. Study labels and representative accession number of sequences carrying the indicated spike mutations are listed. Mutations are located on the spike protein structure based on the PDB:5X59 file. Domains are color for NTD as green, RBD as orange and S2 as blue. **(E)** Mutations on the spike surface for each domain are colored according to sampling years using Chimera software.

compared with earlier strains [18]. A recent study showed MERS-CoV sequences collected from camels in Saudi Arabia in 2023–2024 had emerged from previous clade B lineage 5 strains and had acquired unique genetic features [19]. Spike mutations in MERS-CoV have been shown to modulate viral replication in-vitro and pathogenicity in a hDPP4-knockin mouse model [20]. The spike I529T mutation (reported to arise during human outbreak in Korea in 2015) has been shown to have lower cell entry, reduced viral replication and an attenuated phenotype in the mouse model [20,21]. Similarly, spike mutations in an African clade C strain from Burkina Faso have also demonstrated reduced viral replication in human respiratory tract tissues [22]. These observations suggest that MERS-CoV spike protein can be one major molecular determinant of the virus phenotype in humans.

Here, we hypothesised that mutations in spike protein in more recent clade B MERS-CoV may contribute to reduced pathogenic potential to humans, resulting in reduced numbers of zoonotic MERS in recent years. We analysed MERS-CoV spike protein mutations observed since 2015 and generated spike-packaged pseudoviruses to assess any change of cell entry from representative spike proteins. Using reverse genetics, we rescued recombinant viruses with different chimeric spike protein mutants on the clade B lineage 5 2015/GD01 MERS-CoV backbone and compared the replication kinetics contributed by the spike protein. We also addressed potential mechanistic changes due to the spike mutations in terms of entry pathway preferences and receptor DPP4 binding. We found reduced replication from 2019 and 2023 sequences in multiple cell models. Mechanistic mutations were identified in both RBD region and S2 domain. These data highlight a need for further investigations of spike mutations to understand the potential contribution to the reduced human MERS cases.

## Results

### Spike mutations accumulate during MERS-CoV evolution

We retrieved human and camel MERS-CoV sequences available from Genbank and included newly published sequences collected from camels in Saudi Arabia in 2023–2024 for our analysis (total n = 607) [19]. Phylogenetic analysis showed MERS-CoVs in 2016–2017 had diverged into multiple clade 5 sub-lineages in 2016–2017, but a single sub-lineage dominated from 2018 onwards (Fig 1B). Recent virus sequences from 2023 and 2024 were monophyletically clustered with those from 2019, suggesting that these recent viruses shared a recent common ancestor and a direct evolutionary link. MERS-CoV spike protein is responsible for host cell entry and determines the host-species tropism. Given the limited number of human cases reported after COVID-19, we aimed to investigate whether the spike protein plays a major role explaining the recent changes in epidemiology of MERS-CoV in the Arabian Peninsula. We used the ancestral clade B lineage 5 2015/GD01 strain, a virus representing the 2015 outbreak in Republic of Korea, as the reference of this study. By aligning the spike protein sequences, we observed an average number of 1.7 to 2.9 amino acid substitutions in viruses from 2016 to 2019 sequences, respectively (Fig 1C). The number of amino acid substitutions further increased in 2023 and 2024 sequences, with an average of 8.7 and 9.4 mutations, respectively, relative to 2015/GD01.

The observed spike mutations occurred in spike S1 N-terminal domain (NTD), receptor binding domain (RBD) and spike S2 domains (Fig 1D). Structural analysis revealed most of these mutations were located at the surface of the NTD, with the exception of the A205S mutation (Fig 1E). Similarly, all eight mutations on the RBD were located on the protein surface. The Q833R and Q914K mutation were located the surface of the S2 trimer. The lack of sequence data from 2020-2022 hindered a more sequential tracking of how spike mutations in 2023 emerged.

To investigate the phenotypic effect of these observed mutations, we picked representative sequences from early clade B lineages (2013/B2, 2014/B3 and 2014/B4), 2018 (2018/AF and 2018/T424I), 2019 (2019/LAK and 2019/SV) and 2023 (2023/B5-1, 2023/B5-2, 2023/B5-3 and 2023/B5-4), to compare their cell-entry using pseudotyped viruses expressing representative spike proteins and the effect on viral replication in-vitro using infectious chimeric 2015/GD01 viruses expressing the selected virus spike proteins. The I529T mutation previously identified in 2015 Korean sequences associated with reduced human DPP4 binding and mice pathogenicity was used as an additional control [20,21].

**Impact of MERS-CoV spike mutations on cell-entry**

We generated lentiviral-pseudoviruses expressing different spike proteins and measured cell-entry in Vero and Calu3 cells. It has been shown that MERS-CoV utilizes the endocytic pathway for entry into Vero cells, whereas virus entry into Calu3 cells are mediated by transmembrane protease, serine 2 (TMPRSS2) activation at the cell surface [23]. We first tested early clade B lineage 2, 3 and 4 spike proteins, which contain mainly S2 Q1208H, C1333F and Q833R mutation respectively, and showed they did not show significant changes in cell-entry relative to 2015/GD01 spike (Fig 2A). Similarly, 2018/AF, which contains V109A and L745F mutations, and 2018/T424I, with an T424I mutation at the RBD, also showed no significant difference in virus entry compared with 2015/GD01. Next, we tested spike from 2019 sequences and observed significant reduction of cell-entry in both Vero and Calu3 cells for 2019/LAK and 2019/SV spike (Fig 2B), the former showing greater reduction compared to 2019/SV in both cell types. The extent of reduction of cell entry of 2019/LAK was comparable to the previously reported virus mutation 2015/I529T. Interestingly, the 2019/LAK virus contains a V530A mutation, which is located adjacent to the I529T mutation previously reported during the Korean outbreak. We further identified that the V530A mutation in 2019/LAK and the V1301L mutation in 2019/SV were the main determinant of the cell entry phenotype in both spikes (Fig 2C). The V1301L mutation with a reduced cell entry is located at the transmembrane domain of the spike protein. We did not observe any reduction in cell entry associated with the 2023 sequences, although 2023/B5-3 showed a weak but significant reduction of cell entry of 0.23 $\log_{10}$ RLU (Fig 2D). We harvested the protein lysate from pseudovirus virion supernatant to check the spike cleavage at the S1/S2 site and the packaging of spike into virions (Fig 2E). We found no significant differences in terms of spike cleavage and incorporation.

**In vitro replication kinetics of recombinant viruses with representative spike mutants in comparison to MERS-CoV 2015/GD01**

In order to confirm the data obtained from pseudotyped viruses, we sought to assess the impact of selected virus spike mutations in infectious chimeric 2015/GD01 MERS-CoV expressing the mutant virus spikes. A detailed bio-risk analysis was carried out prior to the initiation of these experiments (see methods). We utilized the Golden Gate assembly strategy to modify the pBAC infectious clone encoding a MERS-CoV 2015/GD01 genome into different pBAC infectious clones encoding chimeric recombinant MERS-CoV 2015/GD01 backbone with different spike variants (Fig 3A-3C, details refer to methods). A total of 7 recombinant viruses were rescued and tested in this study: 2015/GD01 (reference virus), 2015/I529T, 2018/T424I, 2019/LAK, 2019/SV, 2023/B5-3 and 2023/B5-4 (spike mutation as shown in Fig 1D). We observed that 2015/I529T, 2019/LAK and 2019/SV showed smaller plaque size in Vero cells relative to 2015/GD01, but 2018/T424I, 2023/B5-3 and 2023/B5-4 viruses showed relative larger sizes (Fig 3D). As expected, 2015/I529T demonstrated similar smaller plaque morphology compared with 2015/GD01, as previously reported by other studies using a clade A recombinant MERS-CoV/EMC backbone [20,24].

We compared the growth kinetics of these viruses in VeroE6-TMPRSS2 and Calu3 cells by infecting cells at a low multiplicity-of-infection (MOI) of 0.01. In VeroE6-TMPRSS2 cells, 2018/T424I showed similar growth kinetics compared to 2015/GD01, but significant reduction of infectious virus titers was seen in 2019/LAK and 2019/SV at 24 and 48 hours post infection (hpi) (Fig 3E). The 2019/LAK virus showed a reduction of 3.04 and 1.43 $\log_{10}$ TCID$_{50}$/ml at 24 and 48 hpi,

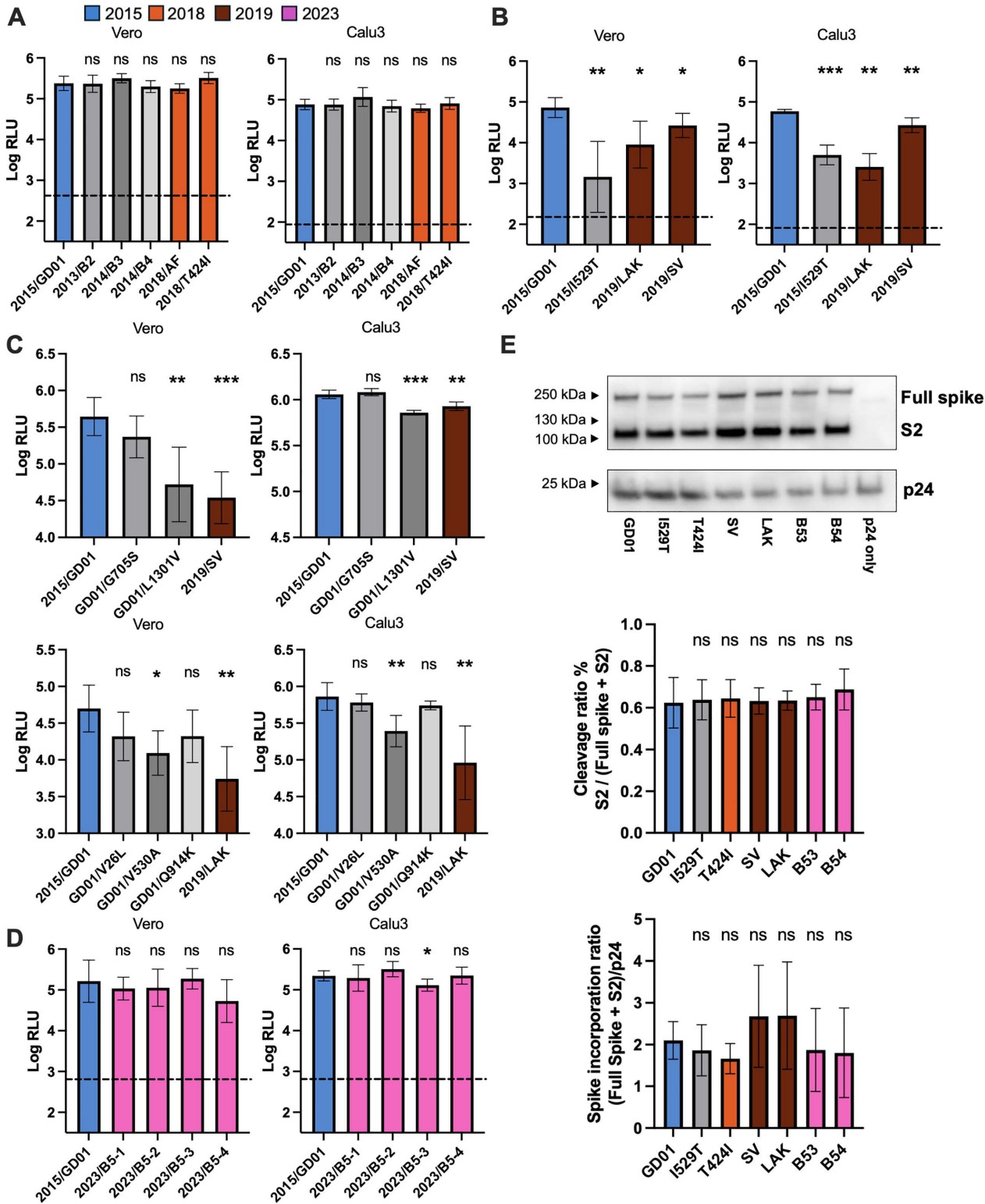

**Fig 2. Cell entry of spike mutations using pseudovirus assay. (A-D)** Cell entry of spike protein expressed lentivector pseudoviruses in Vero and Calu3 cells were measured by relative luciferase units (RLU). Spike expression plasmids encoding **(A)** clade B lineage 5 viruses from 2015 to 2018, **(B)** sequences from 2019, **(C)** individual single mutation from 2019 viruses, and **(D)** sequences from 2023, were co-transfected with third-generation

lentivirus expression plasmids in 293T cells. In **(A)-(D)**, a normalized dose of 7-8 $\log_{10}$ p24 RNA copies per 100ul of pseudovirus supernatant was added to target cells. Dotted horizontal line showed the RLU from "bald" non-spike expressed pseudovirus controls. Assays were performed with five replicates using two independent batch of pseudoviruses. **(E)** Western blot analysis of pseudovirus virions with different spike proteins. Spike proteins was stained by an anti-S2 antibody. The upper band corresponds to a full-length spike, whereas the lower band corresponds to a cleaved S2 protein. Virion quantity was measured by the levels of lentiviral p24 antigen. The data are from n = 4 independent replicate experiments and the blot shown is a representative example. The data are presented as the average ± SD. Statistically significant differences between 2015/GD01 and mutant spikes pseudoviruses were determined by two-sided Student's t tests (* $p < 0.05$, ** $p < 0.01$, *** $p < 0.001$). Color legend: 2015/GD01 (blue), 2018 spikes (orange), 2019 spikes (brown) and 2023 spikes (magenta).

respectively, compared with reference virus 2015/GD01, whereas the 2019/SV showed a smaller reduction of 1.46 and 0.52 $\log_{10}$ $TCID_{50}$/ml at 24 and 48 hpi, respectively. Similar comparative reductions virus growth kinetics were seen in Calu3 cells with 2019/LAK and 2019/SV, although the magnitude of reductions observed was smaller (Fig 3F). Area-under-curve (AUC) analysis showed 2015/I529T, 2019/LAK and 2019/SV produced less infectious virus as compared with 2015/GD01, in both cell cultures. In AUC analysis, we did not observe significant differences in replication kinetics with 2023/B5-3 and 2023/B5-4, compared with 2015/GD01 in both cell types (Fig 3G and 3H). Reduction in replication was seen with 2023/B5-4 in VeroE6-TMPRSS2 cells at 48 hpi (0.49 $\log_{10}$ $TCID_{50}$/ml) and with 2023/B5-3 in Calu3 cells at 72 hpi respectively (0.31 $\log_{10}$ $TCID_{50}$/ml).

To extend the phenotypic comparison, we further tested the replication of recombinant viruses with spikes from 2019 (LAK and SV) and 2023 (B5-3 and B5-4) in human primary nasal (Fig 4A) and alveolar epithelial cells (AECs) (Fig 4B). In nasal epithelial cells, 2019/LAK showed significant reduced viral titers at 24 and 48 hpi, whereas 2023/B5-4 only showed a reduction at 24 hpi. AUC analysis showed the 2019/LAK, and to a lesser extent, 2023/B5-4 had reduced replication in nasal cells compared to 2015/GD01. In alveolar epithelial cells, both 2019/LAK and 2023/B5-4 showed reduction of viral titers at 24 and 48 hpi. A modest reduction was observed in 2019/SV at 48 hpi. AUC analysis showed similar overall replication reduction in 2019/LAK and 2023/B5-4, whereas a modest reduction was seen in 2019/SV. The minor reduction trend in 2023/B5-3 did not reach statistical significance (p = 0.058).

We measured the RNA expression of DPP4 in primary AECs and Calu3 cells and found a substantial reduction of DPP4 expression, coupled with a minor decreased TMPRSS2 and a minor increased cathepsin L (CTSL) expressions (S1 Fig). These findings may explain differences in observed phenotypes of spike mutants in different cell types.

To confirm the phenotypes observed, we performed competition fitness assays by infecting a mixture of 2015/GD01 and spike mutants at ratios of 1:1 and 1:9 in the same culture well of alveolar epithelial cells. The genotypes of virus populations in the inoculum and viral supernatant at 24 and 48 hpi were determined by Sanger sequencing (S2A Fig). Experiments were performed in three independent tissue donors for data reproducibility. For infection with a 1:1 ratio of 2015/GD01 and mutant, we observed mutants 2019/SV, 2019/LAK and 2023/B5-3 were outcompeted by 2015/GD01 in 2 out of 3 donors, whereas 2023/B5-4 were outcompeted in all three donors at 48 hpi (Fig 4C). The variability of the competition phenotype between donors may reflect the individual donor variation. For infection with a 1:9 (GD01 vs mutant) ratio, we observed each spike mutant remained dominant at 48 hpi, except one 2019/SV infected donor showed transient conversion to GD01 (S2B Fig). We believe the short time span of a single-round infection could not allow the more diluted virus to regain its fitness advantage in the competition. Overall, our primary cells infection data showed spike mutants 2019/LAK, 2019/SV and 2023/B5-4 were having consistent lower replication fitness than 2015/GD01.

## Mechanisms of virus entry

We next investigated if any of these spike mutations impact the virus entry pathways of MERS-CoV, specifically the relative role of the cell surface mediated TMPRSS2 entry pathway and the endosomal cathepsin L entry pathway. In SARS-CoV-2, Omicron variant BA.1 showed a switch to the use of the endosomal pathway for cell entry, compared to the ancestral strains which predominantly used the cell surface pathway for cell entry [25]. In the case of Omicron BA.1,

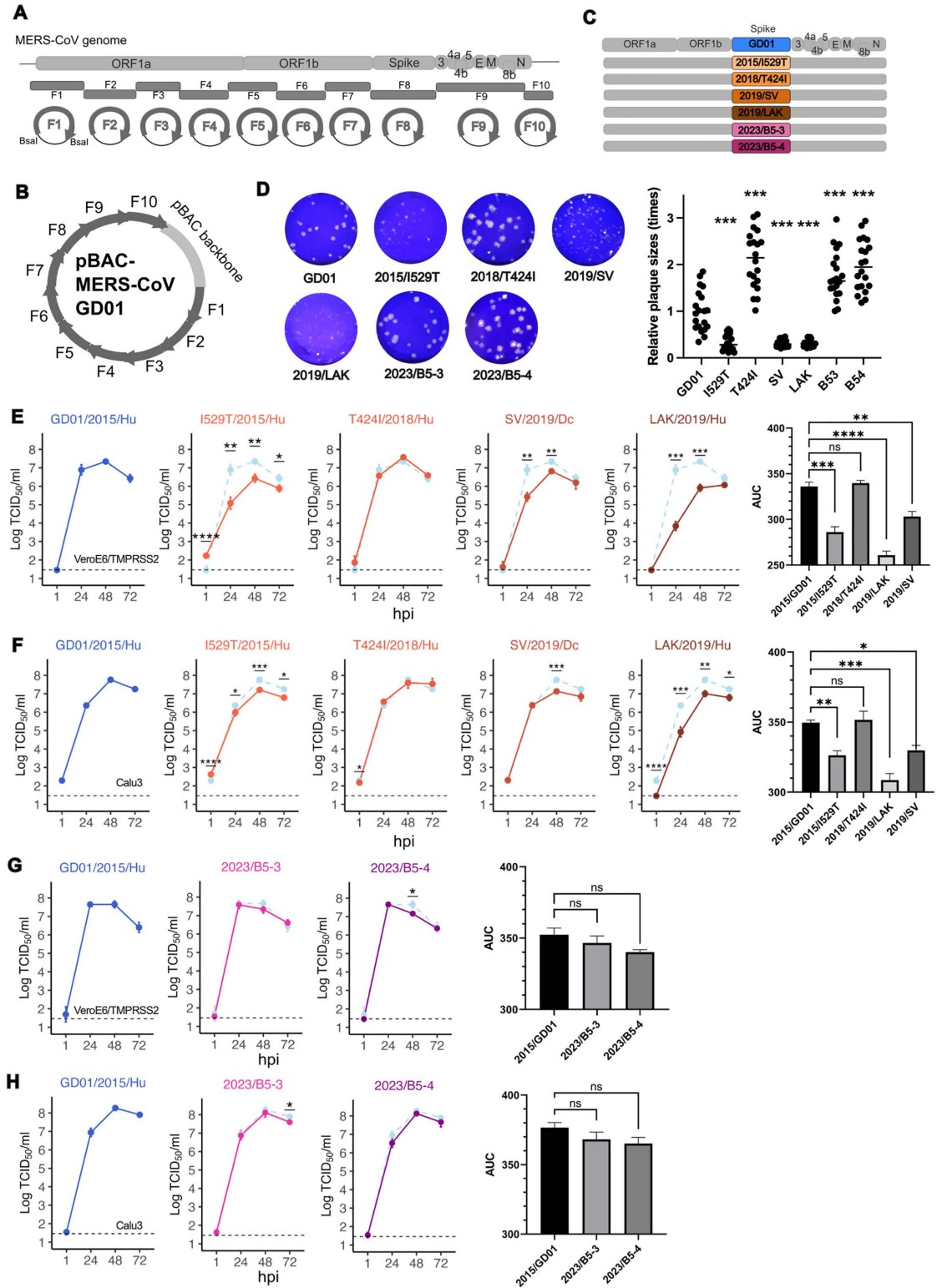

**Fig 3. Replication kinetics of recombinant MERS-CoV/GD01 with chimeric spike variants. (A)** Scheme of the strategy to generate infectious clone encoding recombinant MERS-CoV/GD01 with chimeric spike variants using the Golden Gate assembly. **(B)** A 38kb circular pBAC infectious clone encoding the complete genome of a recombinant MERS-CoV 2015/GD01. **(C)** MERS-CoV genome and its individual genes. Spike genes encoding the mutant

variants were swapped with the original GD01 spike. **(D)** Plaque assay of rescued recombinant MERS-CoV/GD01 viruses in Vero cells. Relative plaque sizes were measured by Fiji. **(E-F)** Replication kinetics of recombinant MERS-CoV/GD01 encoding 2015/I529T, 2018/T424I, 2019/SV and 2019/LAK vs 2015/GD01 in **(E)** VeroE6/TMPRSS2 cells and **(F)** Calu3 cells. **(G-H)** Replication kinetics of recombinant MERS-CoV/GD01 encoding 2023/B5-3 and 2023/B5-4 vs 2015/GD01 in **(G)** VeroE6/TMPRSS2 cells and **(H)** Calu3 cells. **(E-H)** Horizontal dotted line in the line chart represents the limit of detection for the TCID$_{50}$ assay. Figures show the representative data from 2 **(G-H)** to 3 **(E-F)** independent experiments each with three biological replicates. Each dot represents the average ± SD. The bar chart represents the AUC analysis using datapoints from 24, 48 and 72 hpi. Error bars indicate the standard error of the mean. Statistically significant differences between 2015/GD01 and other spikes were determined by two-sided Student's t tests (* $p < 0.05$, ** $p < 0.01$, *** $p < 0.001$; **(E-G)** $p < 0.0001$ ****).

this entry pathway switch was contributed by mutations in the S2 domain. We observed several S2 mutations among our studied MERS-CoV spike sequences: Q914K in 2019/LAK, G705S and L1301V in 2019/SV and 3 conserved S743I, 1179–1180 del among the 2023/B5-1 to B5-4 spikes. We first investigated the cell entry pathway preference using inhibitors targeting either TMPRSS2 (Camostat) or Cathepsin L (E64d) in VeroE6-TMPRSS2 cells, which offer both pathways for cell entry. Using recombinant viruses to infect cells pre-treated with entry inhibitors, we observed that only Camostat effectively inhibited viral replication, starting at 10 µM concentration (Fig 5A). No effective inhibition was observed using the endosomal inhibitor E64d. Similarly, pseudoviruses expressing different MERS-CoV spikes showed the same entry preference via the TMPRSS2 entry pathway in VeroE6-TMPRSS2 cells (Fig 5B). To further confirm how these spikes utilized the TMPRSS2 entry pathway, we repeated the pseudoviruses infection experiments in Caco2 and Huh7 cells. By assessing the RNA expression of DPP4, TMPRSS2 and CTSL, and surface expression of DPP4 and TMPRSS2, we identified Caco2 cells were both high in DPP4 and TMPRSS2 expression, whereas Huh7 cells were high in DPP4, but low in TMPRSS2 expression (Fig 5C). Hence, Caco2 will serve as a model for the TMPRSS2 entry pathway, whereas Huh7 will serve as a endosomal CTSL entry pathway model. We found that all spike mutants, as well as GD01, were inhibited similarly in both pathway models, suggesting no significant alterations of entry pathways among these spikes (Fig 5D). Overall, the data suggest that the clade B MERS-CoVs with spike mutations investigated here retain the preferential usage of TMPRSS2 for cell entry.

### Virus entry and receptor binding with DPP4

The I529T and LAK spike mutants were shown to reduce pseudovirus entry in Vero and Calu3 cells (Fig 2). To assess the impact of hDPP4 on this phenotype, we infected pseudoviruses to 293T cells and 293T cells stably over-expressing hDPP4, and compared the fold-increase in virus entry between the two cell types. We observed that the I529 mutation, previously known to reduce virus receptor binding, and 2019/LAK both showed a ~ 500-fold increase in RLU in hDPP4 overexpressing 293T cells while other spike mutant pseudotypes were minimally affected (Fig 6A). This suggested that the reduced entry of the 2019/LAK spike mutant into 293T cells was hDPP4-dependent.

To further assess the binding potential of each spike mutant, we expressed a panel of recombinant S1 domain (codon 21–741) with a c-terminal fused Fc region of human IgG (S3 Fig). We incubated the S1-Fc recombinant protein in Calu3 cells and measured the binding capacity by flow cytometry (Fig 6B). We found that both I529T and 2019/LAK spike showed reduced binding affinities, compared to the GD01 spike. A minor reduced binding affinity was also seen in 2023/B5-4. The 2019/SV showed comparable binding affinity as GD01, corroborating that the V1301L mutation was the phenotype determinant. Overall, these data support the contention that reduced DPP4 binding was the likely mechanism associated with the reduce replicative fitness of LAK spike mutants in this study.

### Discussion

Our study assessed the phenotypic consequences of spike mutations that were observed to occur within the clade B lineage 5 MERS-CoV in relation to cell entry, viral replication, cell entry pathway preferences and receptor binding. We showed spike mutations observed in the 2019 and 2023 virus sequences contributed a lower replication fitness than

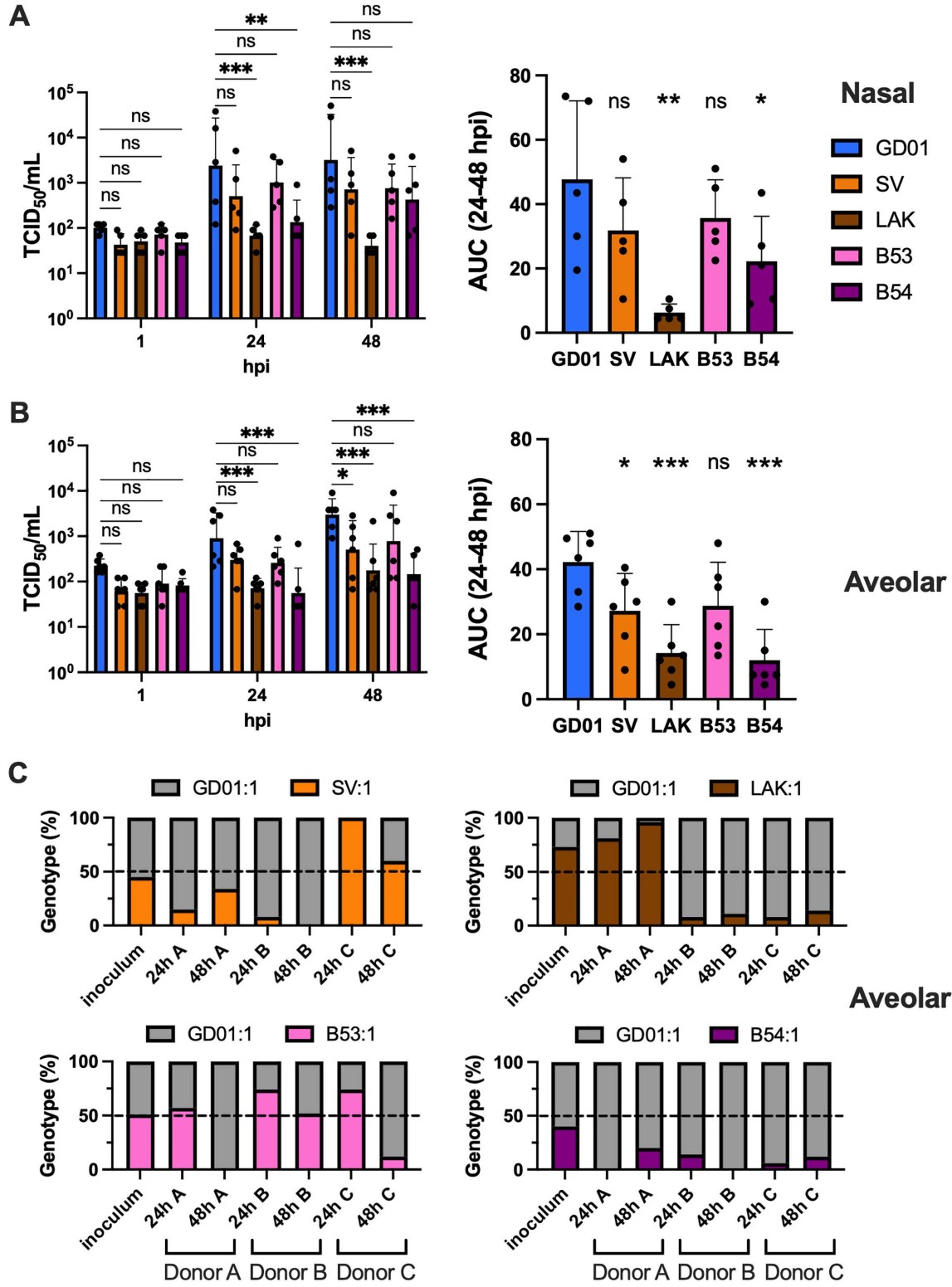

**Fig 4. Replication kinetics and growth competition assay in primary respiratory epithelial cells.** Replication kinetics of recombinant MERS-CoV/ GD01 encoding 2019/SV, 2019/LAK, 2023/B5-3 and 2023/B5-4 vs 2015/GD01 in normal human epithelial cells from nasal **(A)** and alveolar **(B)**. Cells were infected in 5 independent donors for nasal cells and 6 independent donors for alveolar cells, at an MOI of 0.1. The AUC analysis used datapoints

from 24 and 48 hpi. Error bars indicate the SD of the mean. Statistically significant differences between 2015/GD01 and other spikes were determined by two-sided Student's t tests (* p < 0.05, ** p < 0.01, *** p < 0.001). **(C)** Growth competition assay between 2015/GD01 and spike mutants (2019/SV, 2019/LAK, 2023/B5-3 and 2023/B5-4) in normal human alveolar epithelial cells. Experiments were performed in 3 independent donors. An inoculum ratio of 1:1 GD01 vs mutant at a final MOI of 0.1 was used. The proportions of virus in the inoculum, supernatant at 24 and 48 hpi, were determined by sanger sequencing of the RT-PCR amplicons that encode the mutant specific spike mutations for genotyping. Genotype percentages were calculated from the peak height intensities.

the ancestral clade B GD01 strain in 2015. Our findings provide important phenotypic assessment of the role of spike mutations in contemporary circulating MERS-CoV, which will be relevant to the understanding of MERS-CoV emergence potential.

We developed a Golden Gate assembly platform that allowed robust rescue of recombinant MERS-CoV from full-MERS-CoV genome encoding pBAC infectious clones. Similar approaches have been applied from other SARS-CoV-2 studies [26,27]. Starting from synthesis of spike sequences to cloning and rescuing recombinant viruses, the whole procedures were able to be completed within 2 months. The Golden Gate assembly reaction is robust to generate the intact pBAC infectious clone with a high success rate. We believe this technique will be a valuable tool for MERS-CoV research to generate robust virus phenotyping data from new surveillance data and response the emerging threats from the ongoing evolution of MERS-CoV.

We showed that two types of spike mutations that occurred in 2019 MERS-CoV sequences, 2019/LAK and 2019/SV, did contribute to reduced cell entry and virus replication in multiple cell types compared to an early clade B, lineage 5 virus 2015/GD01. On the other hand, the more recent 2023/B5-4 spike, which acquired 10 spike amino acid substitutions, demonstrated reduced viral replication and a less-competitive fitness, only in primary respiratory epithelial cells. We found that human AECs had lower DPP4 expressions and minor variations of TMPRSS2 and CTSL compared to Calu3 cells. The additive changes of these expressions may explain why the reduced virus replication phenotype was only observed in alveolar epithelial cells. It is plausible that the impact of spike mutations on virus-receptor binding affinity manifests in cell types with limiting levels of DPP4 expression. It has also been shown that DPP4 expression is variable in human nasal tissues, which further determines the virus production of MERS-CoV [28]. Our binding affinity data suggest that B53 and B54 remain largely similar to GD01, thus indicating RBD mutations in these spikes may not be crucial in this context. It is possible that mutations at the NTD and S2 could be contributing factors that deserve further investigation. In addition, further experiments perform in differentiated nasal and pulmonary airway cultures can also address potential changes in cell tropism associated with these spike mutations.

We identified a RBD V530A amino acid substitution and a transmembrane domain L1301V amino acid substitution that can contribute to a reduced cell entry in the 2019 spike sequences. The V530A mutation is one codon position adjacent to the I529T mutation, which has previously identified to be associated with reduced cell entry, virus replication and pathogenicity in mice [20,21]. The I529T mutation was reported to have emerged later in the course of human-to-human transmission chains that occurred during the outbreak in the Republic of Korea in 2015 [29]. Interestingly, we found that the L530A mutation at the adjacent codon position (530) also demonstrated reduced DPP4 binding. Both the reduced binding phenotypes at the codon position (529–530) suggest these sites are critical for receptor binding. For 2019/SV, the L1301V mutation located at the transmembrane domain of the spike was responsible for the reduce cell entry. The spike transmembrane domain consists of a hydrophobic domain, which is important for stabilizing the trimeric structure and membrane fusion. Replacing hydrophobic residues into hydrophilic residues in the transmembrane domain showed reduced pseudovirus cell-entry for SARS-CoV spike protein [30]. The leucine to valine residue change may possibly affect the hydrophobicity due to a larger isobutyl group in leucine compared to the isopropyl group in valine. Further experiments using a cell-cell fusion assay may elucidate the mechanism of the L1301V mutation.

While our study highlighted that spike mutations in clade B MERS-CoV can be one factor determining the replication fitness of MERS-CoV, it is important to note that mutations at other parts of the genome other than spike protein may

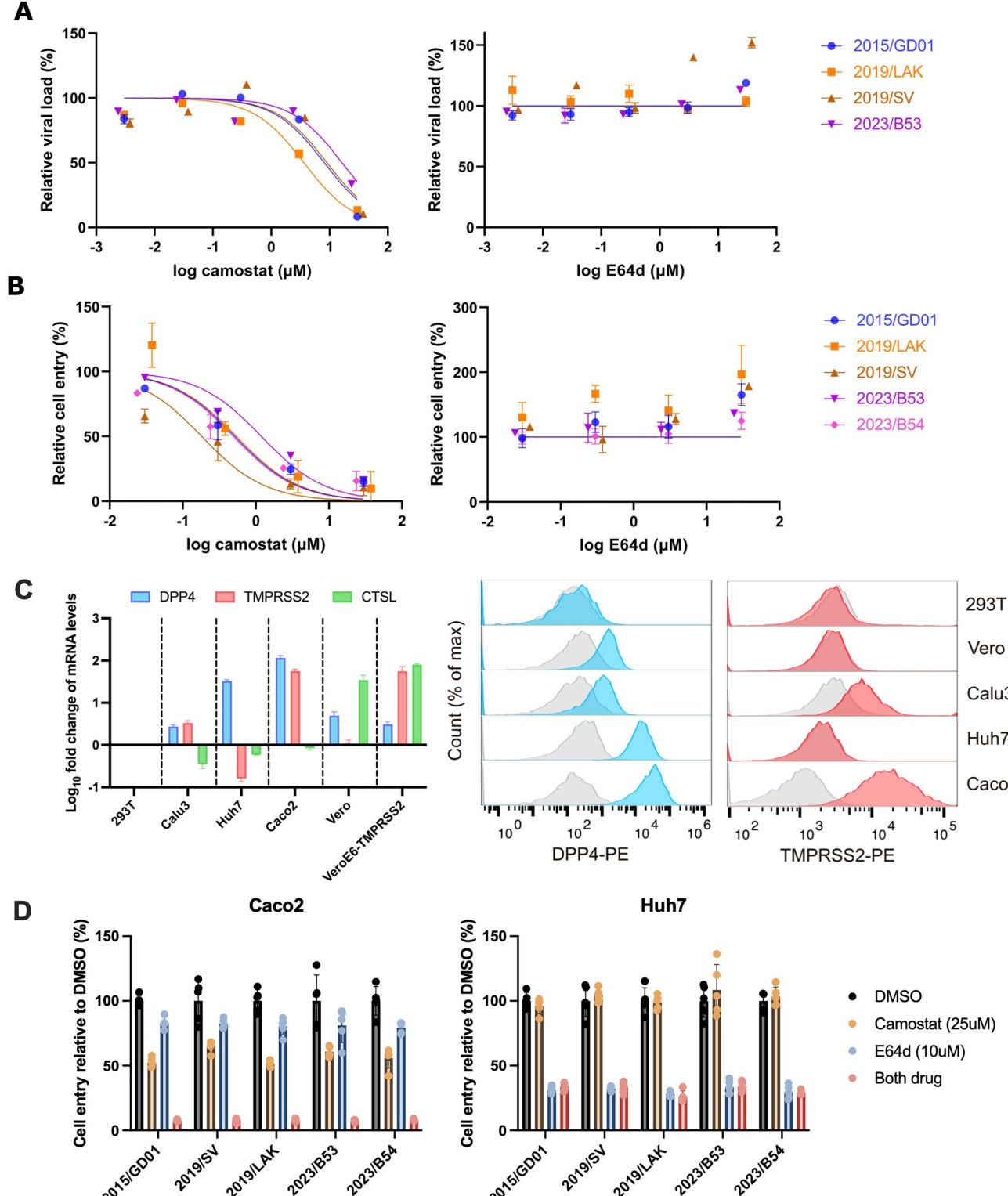

**Fig 5. Entry pathways preference of MERS-CoV spike variants.** Treatment of cells with camostat targeting the cell surface TMPRSS2 pathway and E64d targeting the endocytic cathepsin L pathway was used to block infectious virus entry via each pathway. Experiments with infectious virus **(A)** or pseudovirus **(B)** is shown for the different spike variants in VeroE6/TMPRSS2 cells. **(A)** Relative viral load was measured the RNA copies of MERS-CoV

using the UpE assay in the cell supernatant at 48 hpi. **(B)** Relative cell entry was measured by RLU. **(A-B)** 2015/GD01 was the reference for comparison. Figures show the representative data from 2 independent experiments each with two biological replicates for each experiment each drug dose. An inhibition curve was fitted using the Hill equation with inhibition in Graphpad Prism. **(C)** RT-qPCR to measure RNA expression of DPP4, TMPRSS2 and CTSL gene among difference cell lines. Expressions were normalized by GAPDH gene for each cell line and relative fold changes were compared to 293T cells as reference. Flow cytometry analysis of surface expression of DPP4 and TMPRSS2 among different cell lines. Background levels were determined by isotype controls as indicated in grey. **(D)** Caco2 cells representing the TMPRSS2 entry pathway model and Huh7 cells representing the endosomal entry pathway model were pretreated with Camostat (25 µM), E64d (10 µM) or in both. Pseudoviruses were infected on drug treated cells in quadruplicate wells. Data shown are a representative of two independent repeats.

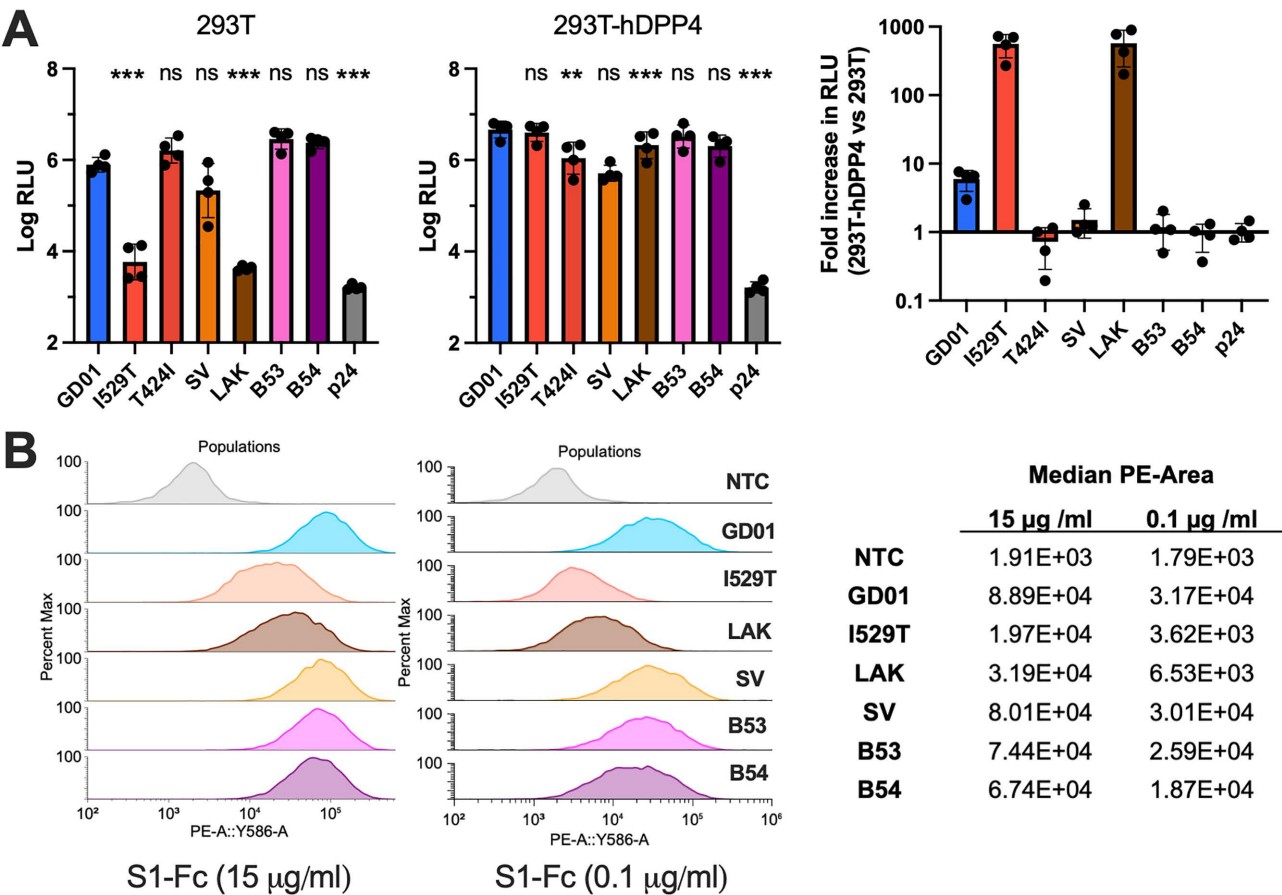

**Fig 6. Functional DPP4 usage and binding affinity assay. (A)** 293T and stable expressing hDPP4 293T cells were infected with different spike expressing pseudoviruses at 37°C. An equal normalized infection dose of 7-8 $\log_{10}$ p24 RNA copies per 100µl was used for both cells and RLUs were measured at 48 hpi. RLU differences between 2015/GD01 and other spikes were determined by two-sided Student's t tests (* $p < 0.05$, ** $p < 0.01$, *** $p < 0.001$). **(B)** FACS analysis of binding of recombinant S1-Fc proteins (at concentrations of 15 and 0.1 µg/ml) to Calu3 cells at 4°C. Protein eluate from non-transfected control (NTC) (grey shading) served as the background signal.

significantly impact virus replication competence and deserve further study. We previously identified a human adaptive mutation, nsp6 L232F, that occurred in the replicase gene increased viral fitness in human cells, and these 2015/GD01 viruses do in fact carry this nsp6 L232F amino acid substitution [31].

The spike V530A and L1301V amino acid substitutions observed in the 2019 MERS-CoV were no longer seen in the 2023 viruses. One reason for this may be due to the purifying selection that result in less fit strains being eliminated in the transmission chains. It is important to note that many of these mutations were driven by selection within the camelid host,

rather than within humans, and thus, the phenotypic consequences of these mutations may or may not be functionally relevant in camelid cells. The impact, if any, of these mutations in camelids requires further experiments to be done in camelid cells or respiratory tissues.

This study has a few limitations. Our investigation was limited to the role of the spike protein. Although spike protein is well recognized as an important determinant of virus tropism and replication competence in coronaviruses, other regions of the genome can also contribute to viral fitness. It has been shown that the emergence of MERS-CoV lineage 5 from earlier lineages is associated with an enhanced IFN antagonism [18]. Thus, we cannot exclude the possibility that reduced zoonotic cases in recent years may indeed be caused by other mechanisms via viral genetic changes elsewhere in the genome, if or if not synergistically with the phenotypic effects contributed by the spike mutations. The 2023 viruses need to be isolated and phenotyped. The paucity of MERS-CoV sequence data in recent years also hinders a systematic analysis of the evolution and phenotype of these viruses. Secondly, our experiments were carried out in cell lines and undifferentiated respiratory epithelial cells. Further experiments in ex vivo human respiratory tissues would be desirable to extend these findings.

In conclusion, spike mutations in 2019 and recent 2023 sequences showed evidence of reduced replication and viral fitness. Our data add novel perspectives to understand the reduced numbers of MERS cases observed in recent years. Other potential explanations for the reduction in human MERS cases also need to be explored and these include the possibility of cross-immunity from COVID-19 infection or vaccination. Although COVID-19 infected or vaccinated individuals have no cross-neutralizing antibody to MERS-CoV, we had previously reported evidence that SARS-CoV-2 infection or vaccination does elicit spike S2 domain binding antibodies [32]. Such antibodies may contribute to some cross-immunity via antibody-dependent cell cytotoxicity or other functional pathways. Separately, cross-reactive T cell immunity may also potentially provide some cross-protection. These possibilities deserve investigation. While public health and social measures would be expected to reduce incidence of most respiratory virus diseases, including MERS, these are no longer in place post 2022 and cannot explain the continued low level of reported MERS cases. Levels of testing for MERS-CoV in many countries in the Arabian Peninsula remains high and lack of testing is unlikely to explain that the recent reduced MERS cases. Our study highlights that MERS-CoV continues to evolve with unpredictable consequences for pandemic risk. Intensive genetic and phenotypic surveillance of MERS-CoV is essential to understand the ongoing changes of the zoonotic potential of these viruses.

## Methods

### Phylogenetic analysis and spike mutation analysis

We analyzed 607 MERS-CoV genomes from Genbank with complete spike gene sequences excluding those arising for cell cultured isolates. The dataset was accessed on 2025 Jan7 and included recent sequences published by Hassan et al [19]. Sequences were aligned by MAFFT and subsequently analyzed to construct a PhyML tree using IQ-tree v2.1.4 using the standard substitution model selection [33]. Spike amino acid mutations against the MERS-CoV 2015/GD01 (accession number: KT006149.2) were counted from the alignment of spike protein sequences using MEGA v10.2.

### Pseudovirus production and infection

HEK293T (ATCC) cells were transfected with a plasmid mix of 850 ng of a pcDNA3.1 expression vector encoding a human codon-optimized MERS-CoV spike protein, 2.5 μg a lentiviral vector backbone encoding firefly luciferase and ZsGreen (Addgene plasmid #164432), 550 ng of a pHDM vector encoding HIV-1 GagPol (Addgene plasmid #164441), 550 ng of a pRC vector encoding HIV-1 Rev protein (Addgene plasmid #164443)and 550 ng of a pHDM vector expressing HIV-1 Tat protein (Addgene plasmid #164442) per $2 \times 10^5$ cells using Trans-IT (Mirus Bio) as transfection reagent. After 48 hours post-transfection, cell supernatant containing pseudoviruses was spun to remove dead cells for 5 mins at 1000g and was aliquoted and store at 80°C until use.

Pseudoviruses were quantified using a RT-qPCR strategy as described by Zhang et al with some modifications [34]. Pseudovirus-containing supernatants were RNA extracted and the RNAs were digested with DNase I according to the manufacturer's protocol to remove any leftover plasmids. DNase-digested RNAs were reverse transcribed using Takara Perfect RT kit. Genomic copies of pseudoviruses were measured by a SYBR-green based qPCR reaction using a pair of primers targeting the luciferase gene (Fwd: 5'-CTTCGAGGAGGAGCTATTCTTG-3' and Rev: 5'-GTCGTACTTGTCGAT GAGAGTG-3') and calculated from a standard curve with known copies of the lentiviral vector plasmid.

To infect target cells, $1 \times 10^8$ genomic copies per 100 µl of pseudovirus was added to pre-seeded Vero and Calu3 cells incubated with 10% DMEM medium. After 48h of incubation in a $CO_2$ incubator at 37°C, cell supernatants were removed and 30 µl of 1x lysis buffer were added for cell lysis for 5 mins. 100 µl of luciferase substrate were added to lysed cells and the luciferase activity was measured using the Perkin Elmer MicroBeta 2 instrument.

## MERS-CoV reverse genetics using pBAC infectious clones

To generate recombinant viruses with different spike variants, we utilized the Golden Gate assembly strategy to induce the desired mutations into a pBAC infectious clone encoding the MERS-CoV 2015/GD01 (obtained from Prof. Jincun Zhao, Guangzhou Medical University). We divided the MERS-CoV genomic region into 10 consecutive fragments and cloned each fragment separately into a pUC high copy number vector. Genomic positions of the fragments were listed in the supporting information (S1 Table). Since the MERS-CoV genome contains 4 internal BsaI cut-site, synonymous mutations were induced into the fragments to abolish these cutsites for the assembly reaction (2438 nt C>A in F1, 8144nt C>A, 11681nt C>A and 11687nt A>G in F4, 21695nt C>T in F8 and 29384nt C>T in F10 respectively). Spike mutations were either induced into the pUC-F8 plasmid, that encodes a full-length spike gene using a site-directed mutagenesis kit (Agilent) or via gene synthesis (Genewiz). To prepare the assembly reaction, 75 ng of each plasmid (F1 to F10 plasmids, plus a pUC vector encoding the pBAC backbone) were added with 2 µl of T4 DNA Ligase Buffer (10X) and 2 µl of NEBridge Golden Gate Enzyme Mix (NEB). The reaction was put into a thermal reaction of 30 cycles of 37°C for 5 mins, followed by 16°C for 5 mins and ended with a step of 60°C for 5 mins. 1 µl of the reaction was electroporated into 25ul of NEB 10-beta Electrocompetent cells using the BTX ECM 630 and Bio-Rad GenePulser electroporators with a setting of 2.0 kV, 200 Omega, and 25 µF. Transformed cells were added with 950 µl of SOC medium and shaken for 37°C for 60 mins. 100 µl of cells were spread on chlorophenol LB agar plates to screen for positive clones. Colonies were picked and screened by PCR with primers flanking the cut-sites to confirm the ligation. pBAC from successful clones were extracted using NucleoBond Xtra Midi prep kit (Macherey-Nagel). Sequence identity of the pBACs were confirmed by NGS on iSeq 100 System (Illumina).

Rescue of recombinant MERS-CoVs from the infectious clones were performed as described previously [31]. Briefly, the pBAC was transfected to Vero cells using lipofectamine 2000 (ThermoFisher). 6 hours later, the transfection mix was removed and replaced with 2% FBS supplemented DMEM medium. After 72h post transfection, the supernatant was passage into VeroE6/TMPRSS2 cells and incubated for another 48h. The presence of CPE will indicate a successful rescue and the supernatant will be kept as P1 for a further next-round passage to generate the experimental stock. All virus stocks were sequenced by NGS, quantified for infectious viral titers using $TCID_{50}$ assay in VeroE6/TMPRSS2 cells and aliquoted for storage at -80°C until use. Transfection of infectious clones and all experiments involved with rescued recombinant viruses were performed in a BSL3 facility in the University of Hong Kong.

## MERS-CoV infection and virus titration $TCID_{50}$ assay

VeroE6/TMPRSS2 (kindly provided from Prof. Peter Cheung, Chinese University of Hong Kong) and Calu3 (ATCC) cells were seeded in 24-well plates one day prior with complete growth medium. Prior infection, cell medium was removed and washed with PBS once. Virus stocks diluted with 0% FBS DMEM medium to a dose of MOI 0.01 were incubated with cells for 1 hour at 37°C. After the incubation, virus inoculum was removed and cells were replenished with 2% FBS DMEM

medium. Supernatants were harvested at 1, 24, 48 and 72hpi for virus titration. $TCID_{50}$ assay was performed in VeroE6/TMPRSS2 cells. Cytopathic effects (CPE) from infected cells were measured at day 5 post infection.

### Isolation of human nasal and alveolar epithelial cells

Nasal samples were collected from subjects without respiratory viral symptoms using a 3-mm bronchial cytology brush (Diagmed Healthcare) to obtain nasal brushings from the nostril. Specimens were placed in BEGM (Lonza) with Primocin (100 μg/ml; Invivogen), centrifuged, and incubated in 0.15% pronase at 37°C for 15–30 min. Digestion was stopped with 10% FBS in BEGM, and cells were washed, counted, and seeded into collagen-coated flasks, maintained at 37°C in a humidified 95% air/5% $CO_2$ incubator.

Primary AECs were isolated from non-malignant lung tissue as previously described [35]. Lung specimens were obtained from patients undergoing lung resection at the Division of Cardiothoracic Surgery, Department of Surgery, Queen Mary Hospital, Hong Kong, China. Bronchi were removed, and lung tissue was minced, washed, enzymatically digested, and depleted of $CD14^+$ macrophages. Cells were cultured in Small Airway Growth Medium (SAGM; Lonza, USA) at 37°C in a humidified incubator with 5% $CO_2$. The study was approved by the Institutional Review Board of the University of Hong Kong and the Hospital Authority Hong Kong West Cluster (UW 24–447 and UW 19–178).

### Primary respiratory epithelial cells infection and competition assay

Virus replication competence was assessed in human primary nasal epithelial and AECs. Cells were infected with virus inocula of MERS-CoV at MOI 0.1. Cultures were infected at 37°C for 1 hour, washed with phosphate-buffered saline and replenished with BEGM for nasal epithelial cells or SAGM for AECs. Culture supernatants were harvested at 1, 24 and 48 hpi and tested for viral titers using TCID50 assays.

To perform competition assay, 2015/GD01 virus was mixed with each mutant virus in ratios of 1:1 and 1:9 (GD01:mutant) infectious titers and infected on primary AECs at a final MOI of 0.1. Each competition setup was infected on three independent donors. Viral RNA was extracted from each inoculum and from supernatant harvested at 24 and 48 hpi. RNAs were reverse transcribed using random hexamers and amplified to obtain PCR amplicons that encode the spike mutations region using specific primers (F: 5'-ccagcagcaattgctagcaactg-3', R: 5'-cgaggtgtgagagtactaggtg-3'). Amplicons were gel purified and sent for Sanger sequencing. Peak heights of the mutation sites were measured using Snapgene software.

### MERS-CoV spike entry pathway analysis using drug inhibitors

Entry inhibitors Camostat (Tocris, Cat.No. 3193) and E64d (Tocris, Cat. No. 4545) were serially diluted with 2% FBS DMEM medium and incubated on to VeroE6/TMPRSS2 cells for 2 hours prior infection. For live virus infection, a virus dose of MOI 0.01 was added to cells on top of the drug-containing medium. Supernatant was harvested at 48hpi and extracted for viral RNA for qPCR quantification using an UpE assay. For pseudovirus infection, $1 \times 10^8$ genomic copies per 100 μl of pseudoviruses were added to drug-treated cells and the RLU was measured at 48hpi.

### Western blotting

To check the cleavage and incorporation of spike protein in pseudoviruses, virions were concentrated by centrifugation at 14000 rpm, at 4°C for 90 minutes. After removing the supernatant, the virions were lysed in 20 μl of 1x loading buffer, and subsequently boiled at 95°C for 10 minutes. Virion lysates were run on a 4–12% gradient Bis-Tris denaturing polyacrylamide gel. Spike proteins were stained by primary anti-MERS-CoV S2 antibody (40070-T62, Sino Biological), followed by anti-Rabbit IgG secondary (#7074, Cell Signaling). p24 proteins were stained by primary anti-HIV-1 p24 antibody (sc-65918, Santa Cruz) and subsequently anti-mouse IgG secondary (#7076, Cell Signaling). Antigen bands were visualized by adding chemiluminescent HRP substrate. Band intensities were quantified by software Fiji.

## Functional DPP4 binding assay

To generate stable hDPP4 expressing 293T cells, 293T cells were transfected with a lentiviral vector encoding a CMV promoter driven cds of human DPP4 (aa 1–765) and a 3' downstream hygromycin resistance gene. 48 hours post transfection, cells were selected with 200 µg/ml of hygromycin B for 2 weeks. Surviving drug-resistant cells were trypsinized and a monoclonal cell population was isolated by limiting dilution. DPP4 expression from isolated colonies were validated by flow cytometry using a PE-conjugate anti-DPP4 antibody (10688-MM05-P, Sino Biological) (S4 Fig).

Equal dose of pseudoviruses were infected on 293T and 293T-hDPP4 cells, seeded at 30000 cells per 96-well, at 37°C as described above. RLU were measured and the fold change increase between cell types were calculated.

To construct the spike S1 domain expression plasmid, the S1 domain cds (codon 21–741) from the codon-optimized spike plasmid was cloned into a pCMV expression plasmid (CV010, Sino Biological) at a site flanked by the signal peptide and human Fc IgG1 gene in the same ORF. To express recombinant S1-Fc proteins, Freestyle 293-F cells in serum-free expression medium were transfected with the pCMV-S1-Fc plasmid using PEI. After 6 days post transfection, cells were spun down to harvest the supernatant containing the S1-Fc protein. To purify the S1-Fc protein, supernatants were passed through a column containing protein A/G agarose, which binds to the Fc region of the protein. Eluted S1-Fc proteins were further spun in a 50 kDa filter column (Amicon) to resuspend the protein in PBS. Protein concentrations were measured by BCA assay. Quality of S1-Fc protein were validated by western blotting (S3 Fig).

FACS binding assay was performed in Calu3 cells by incubating $1 \times 10^5$ cells with 15 µg/ml or 0.1 µg/ml of S1–Fc at 4°C followed by incubation with PE-conjugated goat-anti-human IgG antibody and analysis by flow cytometry.

## Biosafety assessment

We completed an independent risk assessment of the genetic modification of MERS-CoV in this study in the Safety Office, the University of Hong Kong, prior the start of any virus rescue experiments. The spike mutations of 2015/I529T, 2019/LAK, 2015/SV, 2018/T424I, 2023/B5-4 and 2023/B5-3 introduced into the spike protein of MERS-CoV 2015/GD01 and rescued as infectious virus clones were all previously reported to be present in MERS-CoV in the field and thus not novel mutations. Furthermore, each of these spike mutants was previously assessed in this study for virus entry using virus pseudotype experiments which demonstrated that the mutants did not exhibit any increase in virus entry, providing further assurance that these mutations were not likely to enhance virus replication. All laboratory studies with infectious MERS-CoV were carried out in the bio-safety level 3 laboratory at The University of Hong Kong. Virus pseudotype experiments were carried out in bio-safety level 2 laboratory containment, as these are not capable of sustained virus replication.

## Supporting information

**S1 Fig. RNA expression of DPP4, TMPRSS2 and CTSL in primary alveolar epithelial cells.** Three independent donors of alveolar epithelial cells were extracted for cellular RNA. RNAs were reverse transcribed and quantified for expression of DPP4, TMPRSS2 and CTSL. Expression levels were compared by ΔΔCt method across samples with Calu3 cells as reference. Error bars indicate mean with SD.
(TIFF)

**S2 Fig. Growth competition assay in primary alveolar epithelial cells.** (A) Peak information from sanger sequencing of inoculum genotypes. The specific mutation for genotyping of each spike mutant is indicated, and the corresponding nucleotide is marked as asterisk (*). (B) Genotyping result for inoculum at ratio of 9:1 of spike mutant vs 2015/GD01. The 24 timepoint of donor C in B54 did not yield successful PCR amplification.
(TIFF)

**S3 Fig. Expression of recombinant S1 domain (codon 21–741) with a c-terminal fused Fc region of human IgG.** (A) 1ug of purified S1-Fc proteins were loaded on a denaturing 4–12% Bis-Tris SDS-PAGE. The gel was stained with

Coomassie Brilliant Blue R-250 staining solution. Position and sizes of the ladder are indicated. Non-transfected control (NTC) was the negative control to validate no unspecific Fc proteins were purified from the Protein A/G column. (B) Western blot of S1-Fc proteins on a denaturing 4–12% Bis-Tris SDS-PAGE. S1-Fc proteins were stained by primary MERS-CoV Spike Antibody (40069-T62, Sino Biological) and secondary anti-rabbit IgG. The S1-Fc band shared the same position as in the Coomassie Blue staining.
(TIFF)

**S4 Fig. Flow cytometry analysis of DPP4 expression in stable expressing hDPP4 293T cells.** 293T and 293T-hDPP4 cells were stained by PE-conjugate anti-DPP4 antibody (10688-MM05-P, Sino Biological) (Color in Green). Mouse IgG1 kappa isotype PE served as background control (14-4714-82, Thermo) (Color in Blue).
(TIFF)

**S1 Table. Genomic positions of the fragments used in the Golden Gate reaction.** The genome position used is based on MERS-CoV reference genome (Accession: JX869059.2).
(XLSX)

## Acknowledgments

We thank Dr. Mohamed ElGhazaly for the technical advice in the pseudovirus virion western blotting. We thank Jinlin Wang for the technical advice in the expression of recombinant S1-Fc protein. We thank Rui Chen (Centre for Immunology & Infection), Audrey P Y Lai (School of Public Health, HKU) and John K C Li for the technical support.

## Author contributions

**Conceptualization:** Ray T. Y. So, Malik Peiris.

**Data curation:** Ray T. Y. So.

**Formal analysis:** Ray T. Y. So, Kenrie P. Y. Hui.

**Funding acquisition:** Malik Peiris.

**Investigation:** Ray T. Y. So, John C. W. Ho, Kaman K. M. Lau.

**Methodology:** Ray T. Y. So, Kenrie P. Y. Hui, Michael C. W. Chan.

**Project administration:** Kenrie P. Y. Hui.

**Resources:** Ziqi Zhou, Michael C. W. Chan.

**Supervision:** Leo L. M. Poon, Malik Peiris.

**Writing – original draft:** Ray T. Y. So.

**Writing – review & editing:** Ray T. Y. So, Malik Peiris.

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
