## [Decision Letter · Decision Letter 0]

10 Sep 2025

The impact of clade B lineage 5 MERS coronaviruses spike mutations from 2015 to 2023 on virus entry and replication competence

PLOS Pathogens

Dear Dr. Peiris

Thank you for submitting your manuscript to PLOS Pathogens. After careful consideration, we and the reviewers feel that it has merit but does not fully meet PLOS Pathogens's publication criteria as it currently stands. Therefore, we invite you to submit a revised version of the manuscript that addresses the points raised during the review process.

Please submit your revised manuscript within 60 days Nov 09 2025 11:59PM. If you will need more time than this to complete your revisions, please reply to this message or contact the journal office at plospathogens@plos.org. Please include the following items when submitting your revised manuscript:

We look forward to receiving your revised manuscript.

Kind regards,

Simon Anthony, DPhil

Guest Editor

PLOS Pathogens

Michael Letko

Section Editor

PLOS Pathogens

Editor-in-Chief

PLOS Pathogens

orcid.org/0000-0003-2946-9497

Editor-in-Chief

PLOS Pathogens

orcid.org/0000-0002-7699-2064

**Journal Requirements:**

At this stage, the following Authors/Authors require contributions: Ray T Y So, Kaman K M Lau, Ziqi Zhou, Leo L M Poon, and Malik Peiris. Please ensure that the full contributions of each author are acknowledged in the "Add/Edit/Remove Authors" section of our submission form.

- TM on page: 9.

4) Please amend your detailed Financial Disclosure statement. This is published with the article. It must therefore be completed in full sentences and contain the exact wording you wish to be published.

3) If any authors received a salary from any of your funders, please state which authors and which funders.

5) Please provide a completed 'Competing Interests' statement, including any COIs declared by your co-authors. If you have no competing interests to declare, please state "The authors have declared that no competing interests exist". Otherwise please declare all competing interests beginning with the statement "I have read the journal's policy and the authors of this manuscript have the following competing interests:"

**Reviewers' Comments:**

Reviewer's Responses to Questions

**Part I - Summary**

Reviewer #1: The manuscript bySo and colleagues describe phenotypic analyses of MERS-CoV clade B lineage 5 viruses with a particular focus on the spike gene and it's impact on virus fitness. The authors start with the hypothesis that the decline of reported MERS cases since 2018 may be due to phenotypic changes. To address this question, the authors have first generated a set of lentivirus MERS-spike pseudotype viruses contain spike genes representative for MERS-CoV evolution based on the reported MERS-CoV sequences in public databases. As basis for comparison the ancestral clade B lineage 5 2015/GD01 stain was used. Representative sequences were derived from clade B lineages detected in the time period of 2013 to 2023. The lentiviral pseudotypes were used to assess spike-mediated cell entry in Vero and Calu3 cells (for endosomal vs pasmamembrane entry routes). While early clade B lineage 2, 3, and 4 spikes did not show any differences in virus entry compared to 2015/GD01, spikes derived from 2019 sequences showed reduced entry in both, Vero and Calu cells. Notably, spike genes representing more recent variants from 2023 did again not show any difference in entry compared to 2015/GD01. To analyze possible spike-mediated phenotypic changes in MERS-CoV infections, the authors employed reverse genetics to clone the different spike genes into the backbone of 2015/GD01. Again infections were don in Vero and Calu3 cells and overall viral plaque phenotypes and growth kinetics corroborated the lentiviral pseudotype date with viruses contain the 2019 spikes showing slightly reduced growth, while viruses containing the 2023 spikes show similar growth compared to 2015/GD01.

Overall, the described phenotypic analysis of MERS-CoV strains that represent MERS-CoV evolution in camels is of great importance in the context of pandemic preparedness and to assess if phenotypic changes can be observe that may increase or reduce the risk of human infection. However, the study do not use the full spectrum of experimental system that are available. The rather minor phenotypic differences and the exclusive use of cell lines limit the impact.

Specific comments:

1. A logical extension of this study would be the assessment of viral phenotypes in primary culutres, such as nasal, bronchial and alveolar epithelial cultures and organoids. The authors have most of these experimental systems available. It would be of interest if replication kinetics, target cell tropism and possibly host cell responses may differ.

2. To specifically compare viral fitness, competition assay may be useful. In particular if primary airway cultures can be employed.

3. The authors may want to include lack of reporting of MERS cases during the COVID pandemic as one additional potential reason for the decline in reported MERS cases.

Reviewer #2: MERS-CoV is an important emerging coronavirus with zoonotic potential and a high mortality rate in humans. This study seeks to understand the recent decrease in human cases of MERS-CoV by examining spike mutations which have arisen in recent years. They also utilize a Golden Gate cloning strategy for efficient synthesis of MERS-CoV infectious clones, which could be of use in the field. This study provides interesting preliminary data utilizing the sequences of MERS-CoV genomes available and generates many interesting questions to address in the future. There are a few areas where the paper could be strengthened and the authors should be careful with their language on how their findings connect to the recent decrease in human cases. Overall, the study is very well written and clear. The figures are organized and well put together.

Reviewer #3: In the paper, So et al discuss the impact of substitutions that have appeared in strains of MERS on viral entry and replication. They employ both pseudotyping and live virus approaches to assess the contribution of mutations. While the data is interesting, the authors do not provide a comprehensive picture of the effect of the studied substitutions nor mechanism, leaving many open questions.

**Part II – Major Issues: Key Experiments Required for Acceptance**

Reviewer #1: The inclusion of primary airway cultures is important

competition experiments to corroborate growth kinetics results.

Reviewer #2: (No Response)

Reviewer #3: For the all the S studied in using the lentiviral pseudotyping approach, authors should purify and analyze the incorporation/presentation of S on the pseudovirions by immunoblot. They need Spike westerns and functional DPP4 usage experiments for all these mutants to understand processing and S1/S2 status on pseudoparticles.

Concerning the mechanism of entry, the authors investigated if the substitutions detected have an impact on the pathway of entry. This was assessed using Camostat and E64d, inhibitors of TMPRSS2 and Cathepsin L, respectively. In addition to this, the authors could consider comparing cell lines with high and low surface expression of TMPRSS2, such as HuH7 and Caco2/Calu3.

Figure 2. There is no data for some mutations, e.g. V26L or Q914K. Please add a full breakdown of individual point mutants.

Figure 3: The main results from Figure 2 are not taken forward to Figure 3. Why have the authors not made the individual point mutations (e.g. V530A) in live virus?

**Part III – Minor Issues: Editorial and Data Presentation Modifications**

Reviewer #1: see point 3 in part I

Reviewer #2: 1. Abstract line 38-40 and discussion line 325-327: While the data my suggest that the spike mutations are not driving the decrease in human cases, viral entry and replication were only examined in 2 tissue culture cell lines. This is not sufficient to conclude that mutations in the spike are or are not contributing to the number of human cases. The authors should soften their language in these places to reflect this.

2. The amount of spike incorporation into lentivirus pseudotypes should be reported, perhaps via ELISA quantificaion. It is possible that the decrease in entry from the 2019 strains could be due to decreased spike incorporation especially given one of the mutations seen is in the TM.

3. Plaque sizes of infectious clones should be quantified and reported in paper.

4. Figure 3 – please indicate unit of time for X axis in Figure 3 E, F, G, and H in the figure legend.

Reviewer #3: In the discussion, the authors talk about SARS-CoV-2 S2-specific antibodies, and their potential impact for cross-protection against MERS. There is a recent study that shows no cross-immune response against MERS of sera from SARS-CoV-2 infected individuals [1], while paper suggests that anti-S2 antibodies cross react against MERS and its related viruses [2]. They should be mentioned in the discussion

It has been reported that MERS lineage 5 is less susceptible to IFN response. I wonder if replication studies should therefore be performed in Vero cells, which are defective for type-I IFN. I would suggest validating the results in human cell lines permissive to MERS, in addition to Calu3. Importantly, it has been showed that lineage 5 MERS increase in viral replication was dependent on reduced induction of immune genes [3], suggesting that evolution of MERS impacts replication at the level of the innate immune response antagonism, and less at the entry step of the viral cycle. The authors briefly mention this, but I think it should be better and more extensively addressed. Overall, the study is interesting and contributes to our understanding of MERS evolution and its role during entry, but I believe the paper addresses only part of the story, with limited results, and should also present data concerning more detailed unravelling of the replication step of the virus, considering that the authors develop and talk extensively about the Golden Gate approach they adapted to the virus.

For figure 2, it was not clear to me what the different colours of the bars represent. After looking at figure 3, I assumed that they were the different strains. I think the colour code should be mentioned in the legend of the figure.

Minor comments:

What is the rationale for looking at entry in Veros? Please examine by flow to understand DPP4 expression relative to Calu3s? Can the authors find more human cell lines expressing DPP4 to do this work?

Line 133: NTD is repeated twice

Line 179-180: it can be just mentioned in the methods, as well as the following paragraph (181-192) describing the Golden Gate approach to prepare the recombinant virus. At least, the paragraph can be shortened.

Line 377: do not need the reference number of luciferase substrate

Line 386: for the Golden Gate system, I would appreciate if the authors would precise which region of the region is included in each segment, to allow reproducibility from other groups.

Line 449: there is a typo of “capable”

[1] Sun L, Man Q, Zhang H, Xia S, Lu L, Wang X, Xiong L, Jiang S. Strong cross immune responses against sarbecoviruses but not merbecoviruses in SARS-CoV-2 BA.5/BF.7-infected individuals with or without inactivated COVID-19 vaccination. J Infect. 2024 Apr;88(4):106138. doi: 10.1016/j.jinf.2024.106138. Epub 2024 Mar 13. PMID: 38490275.

[2] Sun S, He J, Liu L, Zhu Y, Zhang Q, Qiu Y, Han Y, Xue S, Peng X, Long Y, Lu T, Wu W, Xia A, Zhou Y, Yan Y, Gao Y, Lu L, Sun L, Xie M, Wang Q. Anti-S2 antibodies responsible for the SARS-CoV-2 infection-induced serological cross-reactivity against MERS-CoV and MERS-related coronaviruses. Front Immunol. 2025 Mar 28;16:1541269. doi: 10.3389/fimmu.2025.1541269. PMID: 40226608; PMCID: PMC11985752.

[3] Schroeder S, Mache C, Kleine-Weber H, Corman VM, Muth D, Richter A, Fatykhova D, Memish ZA, Stanifer ML, Boulant S, Gultom M, Dijkman R, Eggeling S, Hocke A, Hippenstiel S, Thiel V, Pöhlmann S, Wolff T, Müller MA, Drosten C. Functional comparison of MERS-coronavirus lineages reveals increased replicative fitness of the recombinant lineage 5. Nat Commun. 2021 Sep 7;12(1):5324. doi: 10.1038/s41467-021-25519-1. PMID: 34493730; PMCID: PMC8423819.

PLOS authors have the option to publish the peer review history of their article (what does this mean? ). If published, this will include your full peer review and any attached files.

**Do you want your identity to be public for this peer review?** For information about this choice, including consent withdrawal, please see our Privacy Policy .

Reviewer #1: No

Reviewer #2: No

Reviewer #3: **Yes:** Dalan Bailey

**Figure resubmission:**

**Reproducibility:**



---

## [Decision Letter · Decision Letter 1]

19 Jan 2026

Dear Prof. Peiris,

We are pleased to inform you that your manuscript 'The impact of clade B lineage 5 MERS coronaviruses spike mutations from 2015 to 2023 on virus entry and replication competence' has been provisionally accepted for publication in PLOS Pathogens.

Best regards,

Michael Letko, PhD

Section Editor

PLOS Pathogens

Michael Letko

Section Editor

PLOS Pathogens

Sumita Bhaduri-McIntosh

Editor-in-Chief

PLOS Pathogens

orcid.org/0000-0003-2946-9497

Michael Malim

Editor-in-Chief

PLOS Pathogens

orcid.org/0000-0002-7699-2064

Reviewer Comments (if any, and for reference):

Reviewer's Responses to Questions

**Part I - Summary**

Reviewer #1: The authors have performed substantial additional work and have appropriately addressed the reviewers' comments.

Reviewer #2: The addition of examining infectivity on primary nasal and alveolar epithelial cells greatly strengths the paper. The demonstration of a reduced replication phenotype in the 2023 spikes and mutations in 2019 and 2023 spikes is very intriguing and opens a number of avenues for future study. I feel that all my comments were adequately addressed and that the new experiments added to the paper increase the impact of the paper to the field. Would recommend acceptance at this time.

Reviewer #3: The authors have really improved the manuscript with their additional experiments and I am happy that they've addressed my concerns. Dalan Bailey

**Part II – Major Issues: Key Experiments Required for Acceptance**

Reviewer #1: (No Response)

Reviewer #2: (No Response)

Reviewer #3: n/a

**Part III – Minor Issues: Editorial and Data Presentation Modifications**

Reviewer #1: (No Response)

Reviewer #2: (No Response)

Reviewer #3: My only comment would be for them to perhaps extend their discussion of why the 2023 viruses have no defect in entry, binding but are still slower to grow in the primary cells. I found this intriguing and underexplored in the discussion, although I dont think more experiments are needed at this stge. Also, for the structural figure in Fig 1 it would be nice perhaps colour the mutations by year. I found it difficult to compare this to the table above.

PLOS authors have the option to publish the peer review history of their article (what does this mean? ). If published, this will include your full peer review and any attached files.

**Do you want your identity to be public for this peer review?** For information about this choice, including consent withdrawal, please see our Privacy Policy .

Reviewer #1: No

Reviewer #2: No

Reviewer #3: **Yes:** Dalan Bailey

---

## [Editor Report · Acceptance letter]

Dear Prof. Peiris,

We are delighted to inform you that your manuscript, "The impact of clade B lineage 5 MERS coronaviruses spike mutations from 2015 to 2023 on virus entry and replication competence," has been formally accepted for publication in PLOS Pathogens.

Best regards,

Sumita Bhaduri-McIntosh

Editor-in-Chief

PLOS Pathogens

orcid.org/0000-0003-2946-9497

Michael Malim

Editor-in-Chief

PLOS Pathogens

orcid.org/0000-0002-7699-2064